# Effects of cleft lip on visual scanning and neural processing of infant faces

**Amanda C. Hahn**[1]*, **Juergen A. Riedelsheimer**[1,2], **Zoë Royer**[1], **Jeffrey Frederick**[1], **Rachael Kee**[1], **Rhiannon Crimmins**[1], **Bernd Huber**[1], **David H. Harris**[1], **Kelly J. Jantzen**[3]

1 Department of Psychology, California Polytechnic Institute Humboldt, Arcata, CA, United States of America, 2 Department of Psychology, Idaho State University, Pocatello, ID, United States of America, 3 Department of Psychology, Western Washington University, Bellingham, WA, United States of America

* amanda.hahn@humboldt.edu

**Data Availability Statement:** All data and analysis code for the results reported is publicly available at https://osf.io/6zm9d/. The data files shared contain

## Abstract

Infant faces readily capture adult attention and elicit enhanced neural processing, likely due to their importance evolutionarily in facilitating bonds with caregivers. Facial malformations have been shown to impact early infant-caregiver interactions negatively. However, it remains unclear how such facial malformations may impact early visual processing. The current study used a combination of eye tracking and electroencephalography (EEG) to investigate adults' early visual processing of infant faces with cleft lip/palate as compared to normal infant faces, as well as the impact cleft palate has on perceived cuteness. The results demonstrated a significant decrease in early visual attention to the eye region for infants with cleft palate, while increased visual attention is registered on the mouth region. Increased neural processing of the cleft palate was evident at the N170 and LPP, suggesting differences in configural processing and affective responses to the faces. Infants with cleft palate were also rated significantly less cute than their healthy counterparts (mean difference = .73, p < .001). These results suggest that infants' faces with cleft lip/palate are processed differently at early visual perception. These processing differences may contribute to several important aspects of development (e.g., joint attention) and may play a vital role in the previously observed difficulties in mother-infant interactions.

## Introduction

A growing number of studies suggest that infant faces are evolutionarily salient stimuli that rapidly capture our attention and elicit enhanced neural processing compared to adult faces [e.g., 1–3]. Individual and dynamic infant facial cues may further act as an essential source of information relevant for the allocation of parental resources [reviewed in 4], and trigger intuitive parental responses that support the development of interpersonal communication [reviewed in 5]. The vast majority of studies suggesting that infants' faces undergo "special" processing have been conducted using typically developing, healthy infant stimuli. However, there is strong evidence that infant facial cues related to health influence caretaking behaviors [e.g., 6–8] Anthropological data indicates that cues of poor health are one of the most common

all raw data analyzed, allowing for full replication of the analyses we present.

**Funding:** The authors received no specific funding for this work.

**Competing interests:** The authors have declared that no competing interests exist.

historical and cross-cultural reasons for infant abandonment [e.g., 9]. Among modern societies, cues of low health remain associated with a lower degree of parental investment [10–12], and prenatal detection of fetal abnormalities can lead to increased rates of termination [see 13]. One study even found that nearly 70% of abandoned children carried some conspicuous appearance flaw that was neither life-threatening nor impactful for intellectual development [14, cited in 15]. There is clear evidence that caretakers have different affective and behavioral responses to infants with physical abnormalities. Given the implications for infant developmental outcomes, further understanding the neurobiological underpinnings of these responses is essential.

Cleft lip/palate (a facial malformation that consistently affects the orofacial and sometimes nasal areas across faces) is a common face abnormality estimated to affect 1 in 700 live births worldwide [16, 17]. Children with cleft lip/palate experience a range of long-term adverse outcomes associated with difficulties in early caregiver interactions [18–21] that can still be apparent as late as 7 years of age [22]. Several laboratory-based studies have indicated that infants with cleft lip/palate elicit different responses than their healthy counterparts, consistently showing that the presence of cleft lip/palate significantly reduces the perceived cuteness of infant faces [15, 23–28], a finding that even extends to the faces of domesticated animals [27]. Because infant cuteness reflects baby-typical features (e.g., large eyes, chubby cheeks, small chin, etc.) that act as an innate releasing mechanism for caretaking responses [29], reduced perceptual cuteness may negatively impact parental responses to these infants. Indeed, studies using a keypress task have demonstrated that adults will exert effort to shorten their exposure to infant faces with cleft lip/palate, potentially indicating an aversive response [15, 27]. In line with these findings, Parsons et al. [30] observed reduced activity in the orbitofrontal cortex, a critical reward-related region of the parental brain, when adults viewed infants with cleft lip/palate compared to healthy infants, suggesting that cleft lip/palate may disrupt the neural responses typically associated with caregiving [3]. Although these studies were primarily conducted using nulliparous individuals, the Parental Care Motivational System (i.e., a suite of psychological mechanisms that evolved to regulate caretaking behavior; [31]) is thought to be activated by infant stimuli in parents and non-parents alike. As such, these negative perceptual and behavioral responses to infants with cleft lip/palate may contribute to the observed lack of maternal responsiveness common when an infant has a facial malformation [19, 32, 33].

The current study combines eye-tracking and electroencephalography (EEG) to investigate how the presence of cleft lip/palate may impact visual attention and neural processing of infants' faces. Humans demonstrate a strong bias toward the eyes when viewing a face [reviewed in 34, 35]. The resulting mutual eye gaze is extremely important for mother-infant interactions [reviewed in 5]–it facilitates joint attention and social bonding and has been linked to the quality of relationships between infants and caregivers [36, 37]. The presence of cleft lip/palate may impact mutual gaze by drawing attention away from the eyes; De Pascalis et al. [38] observed that mothers of infants with cleft lip/palate spend less time looking at their baby's face overall compared to mothers of healthy infants. Similarly, Rayson et al. [28] observed that adults fixated on the mouth region of the faces of infants with cleft lip/palate at the expense of the eyes and that the severity of cleft lip/palate predicted this fixation bias. This attentional shift may negatively impact caretaking responses directed toward the infant [18, 19, 32]. For example, Stone and Potton [39] demonstrated that heightened visual attention to the disfigured area of faces was linked to a negative emotional experience and may contribute to stigmatizing affective responses. Moreover, the impact of cleft lip/palate on mother-infant interactions can have lasting negative consequences including impaired cognitive development [19].

Changes in the visual scanning patterns of infant faces with cleft lip/palate may impact early neural processing of the faces. Huffmeijer et al. [24] investigated neural responses to infant

faces with cleft lip/palate, focusing on the amplitudes of the occipitotemporal N170, P100, and P200 event-related potential (ERP) components typically associated with face processing. ERPs are the average of the EEG data that is time-locked to specific events or stimuli, in this instance the presentation of a face. Task dependent cortical processing is reflected in characteristic features of the ERP waveform, commonly referred to as components (see [40, 41] for an introduction to ERPs). The P100 and P200 components represent early and late stages of visual encoding, respectively [e.g., 42]; both are sensitive to configural information contained in faces, and the P200 is thought to reflect the encoding of the second-order spatial relations of a face in particular [43]. The N170 represents the structural encoding of faces and has been associated with configural processing [44, 45]. The N170 has also recently been linked to both positive and negative aspects of parental behavior [46–48]. Huffmeijer and colleagues found that the N170 and P200 responses were attenuated by the presence of a cleft lip/palate (no effect for P100). The impact of cleft lip/palate on the N170 response, in particular, was associated with the impact of cleft lip/palate on attractiveness ratings given to the faces suggesting these processing differences influence later social/perceptual judgments.

The current study aims to extend previous work investigating responses to cleft lip/palate that has provided evidence for differences in early perceptual and neural responses to infants with facial malformations. We do so by measuring visual and neural responses to upright and inverted healthy and cleft lip/palate infant faces in the same participants. Additionally, we investigate the impact of these facial malformations on both early stages of face processing (N170, P200) important for structural encoding *and* later stages of face processing (the late positive potential, LPP) that reflect affective processing and are important for evaluating the impact of the cleft lip/palate on subsequent visual evaluation. The N170 is generally considered an index of face processing and is impacted by configural manipulations [44, 45]. Disruptions to configural processing, such as face inversion, reliably increase the magnitude of the N170, possibly reflecting the recruitment of additional cognitive mechanisms to process the face [e.g., 49]. Because cleft lip/palate disrupts the normal infant facial configuration, we expect an increase in N170 magnitude compared to healthy infant faces. Inverting the face will disrupt configural processing of both cleft and healthy faces and should result in a similar increase in N170 magnitude for both face types. The P200 is generally thought to reflect the encoding of the second-order spatial relations of a face and is considered to be sensitive to the "typicality" of a face [43]. Because they make infant faces less typical, we expect the presence of a cleft lip/palate to also modulate the amplitude of the P200 ERP component. The LPP is a positive ERP component beginning around 400ms after stimulus onset that is modulated by the motivational relevance of the stimuli [50]. It is evoked by emotionally engaging stimuli [51] and is thought to reflect an enhanced evaluation of salient visual stimuli [52]. Because the LPP tends to be larger for negative stimuli [53] and facial disfigurements have been shown to elicit negative emotional responses from observers [e.g., 54–56], we expect the LPP will be enhanced by cleft lip/palate. Lastly, given that cleft lip/palate represents a deviation from normal facial appearance and is linked to atypical visual engagement, we expect to find differences in the visual scanning of cleft faces compared to healthy, normal faces. We predict adults will spend more time viewing the mouth region (at the cost of eye gaze/contact) for the cleft faces.

## Materials & methods

### Participants

Twenty-one participants (13 female, 6 male, 2 non-binary) were recruited from the student population. Recruitment began May 1st, 2022 and concluded June 1st, 2022; the sample size reflects the maximum number of participants we were able to recruit during this time period

(limited by access to the EEG equipment). Participants ranged in age from 18 to 32 years (mean = 22.4 ± 3.2). None reported having children, but seven reported working with children in some capacity (e.g., daycare; mean reported time around children = 2.44 hrs/wk). The sample was primarily right-handed (N = 20, 1 ambidextrous) and White (N = 17). All reported normal/corrected-to-normal vision. Data loss occurred for 3 participants, resulting in a sample size of 18 for analyses reported below.

## Stimuli

Full-color photographs of infants' faces were collected from an online search, 20 infants with cleft lip/palate and 20 healthy infants. Selection criteria for these images included the infant facing the camera head-on, alert (eyes open), with a neutral expression, and no visible adornments (e.g., hair bows). Images were also required to have bright, uniform lighting and be available in the "large" image designation on google images to ensure that the faces were presented in sufficient definition (i.e., did not appear pixelated). All images were aligned on interpupillary distance and then masked with a white background using Webmorph [57] to remove any background, non-face information; this generated standardized stimuli across the face categories. The faces were presented in full color at 600x800px for all elements of the study. Example face images from each category are shown in Fig 1 (note that composite face exemplars were created to protect the identities depicted in the images used in the actual study).

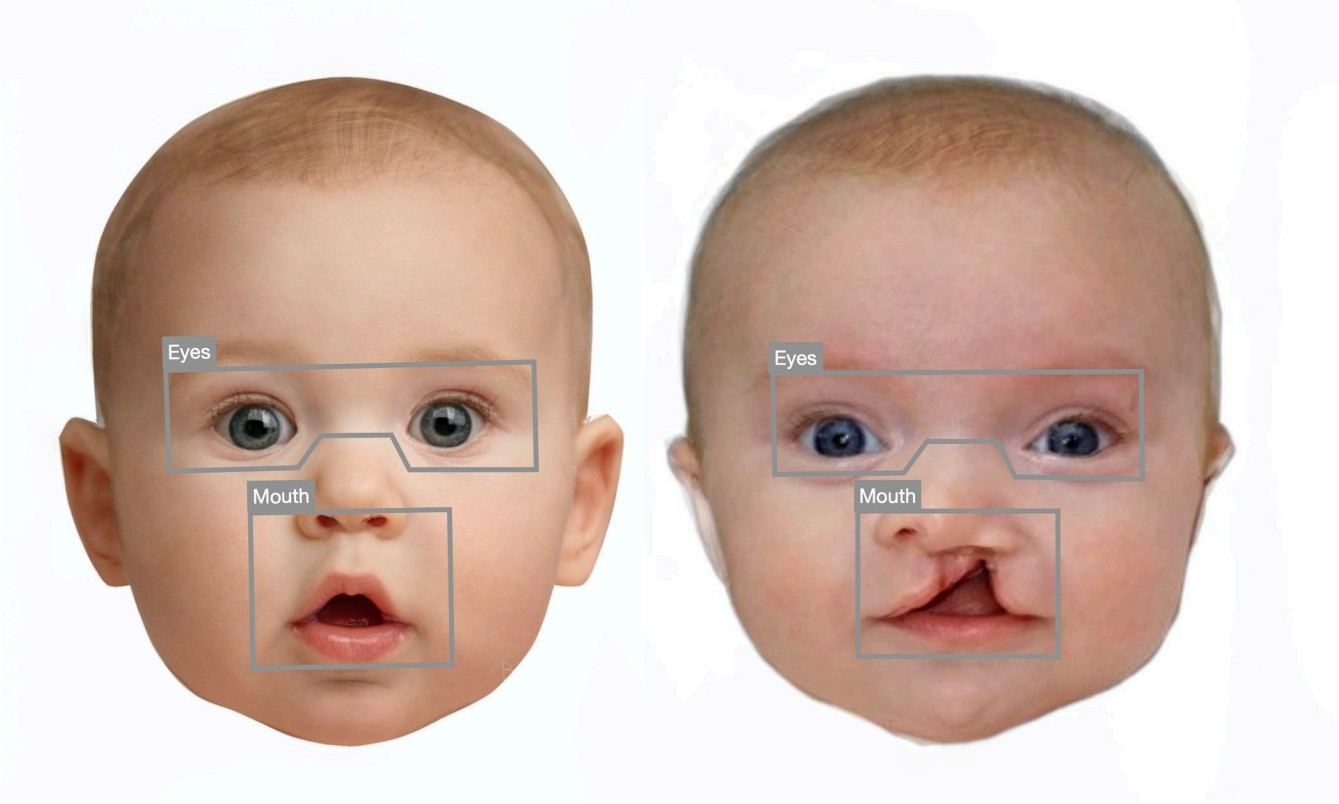

**Fig 1. Representative faces for cleft and normal/healthy infant groups (composite images used here to protect individual identities) illustrating the eye and mouth regions of interest analyzed.**

## Procedure

Participants visited the lab for a 90min study wherein data for the present study and an unrelated study on adult face perception was collected in a counterbalanced order. As data collection took place during the COVID-19 pandemic, they were required to wear a surgical mask that covered the lower half of their face; these masks used behind-the-ear loops to avoid interfering with electrode placement. All participants provided written informed consent. The study was approved by the local IRB (#21–044). All participants were fitted with the EEG equipment prior to beginning the study. Participants then completed two tasks in a fixed order (eye tracking followed by EEG).

Participants were seated on a height-adjustable chair for both tasks and instructed to minimize movements during the recording period. Head movement was limited via a chin rest (Tobii) located 85cm from the computer screen (26" DELL monitor) for the duration of the study. Stimulus presentation and response recording were done using OpenSesame [58].

## Cuteness rating

Following the presentation of each face during the eye tracking portion of the study, participants rated the cuteness of the face they had just seen using a 1 (not cute) to a 5 (very cute) scale. The 40 infant faces were presented in a fully randomized order. Participants advanced to the next task after making their ratings (i.e., there was no time constraint on the rating task).

## Eye tracking

Eye movements were recorded using the GazePoint GP3 infrared eye tracker sampling at 60Hz. The eye tracker was placed below and slightly in front of the monitor, approximately 72 cm from the chin rest. Before beginning the study, a 9-point calibration was performed. Before each trial, a fixation cross appeared at the center of the screen. This was followed by an infant face, displayed for 10 seconds [following 28]. After the 10 second viewing period, the rating task described above appeared on screen before advancing to the next trial.

Two regions of interest (ROIs) were manually selected for each image. The eye ROI encompassed both eyes and was rectangular with the exception of the bottom central border, which followed the bridge of the nose near the nasion. The mouth ROI encompassed the mouth and the lower portion of the nose, extending to the tip of the nose. Although they differed in shape, the area covered by each ROI was similar (see Fig 1).

## EEG acquisition and analysis

During EEG recording, participants were instructed to view the faces presented on the screen (i.e., a passive viewing task). They were asked to focus on the fixation cross that preceded each face and try not to blink when faces were presented. They were also informed that they would be given a break during the viewing task to blink freely and rest. Each of the 40 infant faces was displayed twice in both an upright and inverted orientation for a total of 160 trials. Each trial began with the presentation of a fixation cross at the center of the screen, followed by the presentation of a single face for 500ms. Interstimulus intervals varied randomly between 1000-1500ms.

EEG was continuously recorded from 64 scalp sites, using BioSemi ActiveTwo Ag/AgCl electrodes and hardware (Biosemi, Amsterdam, The Netherlands). Electrodes were placed according to the 10–5 electrode system [59] using a nylon electrode cap. EEG signals were amplified with a bandpass of DC-104 Hz by BioSemi ActiveTwo amplifiers, sampled at 512 Hz. Off-line segmentation and averaging of EEG signals were performed using custom Matlab

scripts (Matlab version 9.12, Mathworks, Inc., Natick, MA, USA) that leveraged the EEGlab toolbox [60]. In a small number of cases, a single channel demonstrated excessive noise and was replaced by a new channel derived by spherical interpolation of the surrounding channels. Signals were referenced to the common average and bandpass filtered between 0.1 and 30 Hz. Individual trial epochs with large artifacts (±500 uV) were removed before applying an independent component analysis (ICA) approach to artifact removal [61]. First, we performed an ICA decomposition on the epoched data of each participant. We then assigned a label to each ICA component using the automated ICLabel plugin for EEGLab [62]. We cleaned the data by removing any components that had less than a 10% chance of being labeled as originating in the brain and a greater than 60% chance of originating from the eye, heart, muscle, or line noise (60Hz). The cleaned trials were averaged separately for each condition.

ERP components were quantified for each participant and condition by averaging the signal within a time and channel montage that captured the spatiotemporal properties of each component. Time windows and channel montages were selected based on previous literature and confirmed by visual inspection of the global field power (for timing) and scalp maps (for montages) of the grand averaged data (see Fig 2). The P100 was defined between 100 to 150 ms and across a set of occipital electrodes, including PO3, POz, PO4, O1, Oz, O2 and Iz. The N170 was defined between 155 and 215 ms for electrodes P7 and P9 in the left hemisphere and P8 and P10 in the right hemisphere. The P200 was evaluated in a window from 220 to 270 ms in

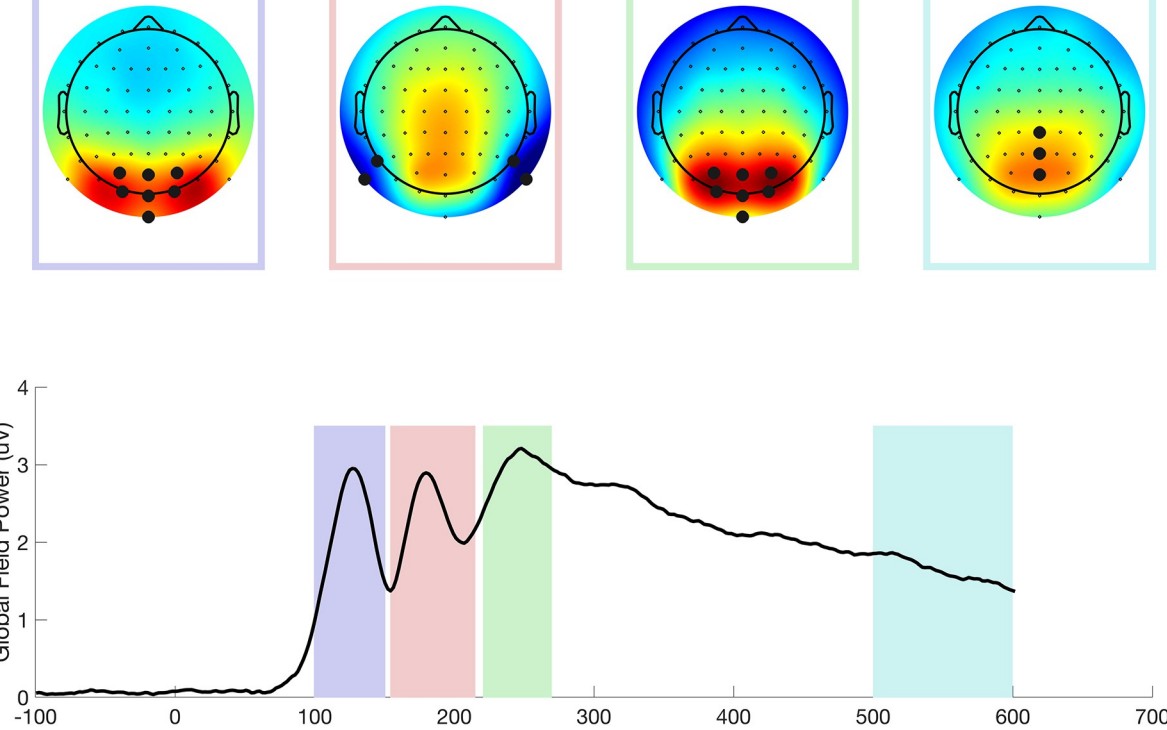

**Fig 2. The bottom panel shows the global field power of the grand average ERP calculated over condition and participant.** The colored regions indicate the temporal windows used to define the P100 (purple), N170 (red), P200 (green), and LPP (aqua) ERP components. The scalp topographies of the averaged amplitude within each window are shown in the insets above. Red is positive, and blue is negative potential. The large dark circles indicated channel locations used for statistical analysis of each ERP component.

the same occipital electrodes. Finally, the LPP was evaluated as between 500–600 ms across a montage of central parietal electrodes (CPz, Pz, P03, P0z, P04).

## Results

All data and analysis code for the results reported below is publicly available at https://osf.io/6zm9d/.

### Cuteness ratings

Cuteness ratings were subjected to a paired-samples t-test to determine the impact of cleft lip/palate on perceived cuteness. Infant faces with cleft palates (M = 2.87 ± 0.56) received significantly lower cuteness ratings than the normal faces (M = 3.60 ± 0.52; $t(17)$ = -6.80, $p < .001$, d = -1.60; See Fig 3).

### Eye tracking

To evaluate the impact of the cleft lip on visual attention, we calculated the time to first fixation (in seconds) and duration of the first fixation (in seconds) for both the eye and mouth regions of interest. Time to first fixation (i.e., the interval between stimulus onset and the first gaze onset) was defined as the onset of the first gaze fixation in each ROI. The duration of first fixation was the dwell time (or length, in seconds) of the first fixation in each ROI. Fixations were determined using Gazepoint's internal fixation filter. We utilized these two measures because the provide insight into how quickly and strongly the mouth region of cleft lip/palate infants capture attention in viewers. These measures were subjected to separate multifactorial repeated-measures ANOVAs with *ROI* (eyes, mouth) and *palate* (cleft, normal) as within-subject factors. To account for the increase in family-wise error rate associated with running statistical tests on the two measures, we set our pre-test alpha at .025 (i.e., .05/2). All posthot tests were Bonferroni corrected using the p.adjust feature in R.

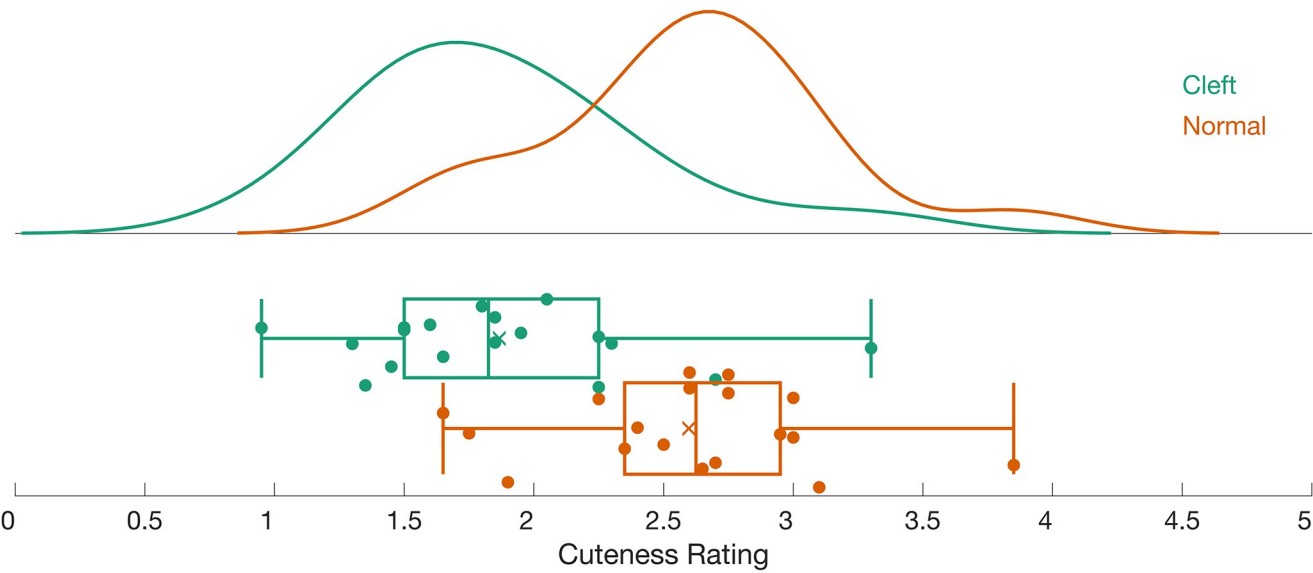

**Fig 3. A raincloud plot of the cuteness ratings.** The histograms show the probability density estimates for rating the cleft (green) and normal (red) palate faces. The individual ratings are shown below as colored circles. The thick vertical line is the mean; the box includes data within the 25th to 75th percentile. The whiskers range from the 2nd to 98th percentile.

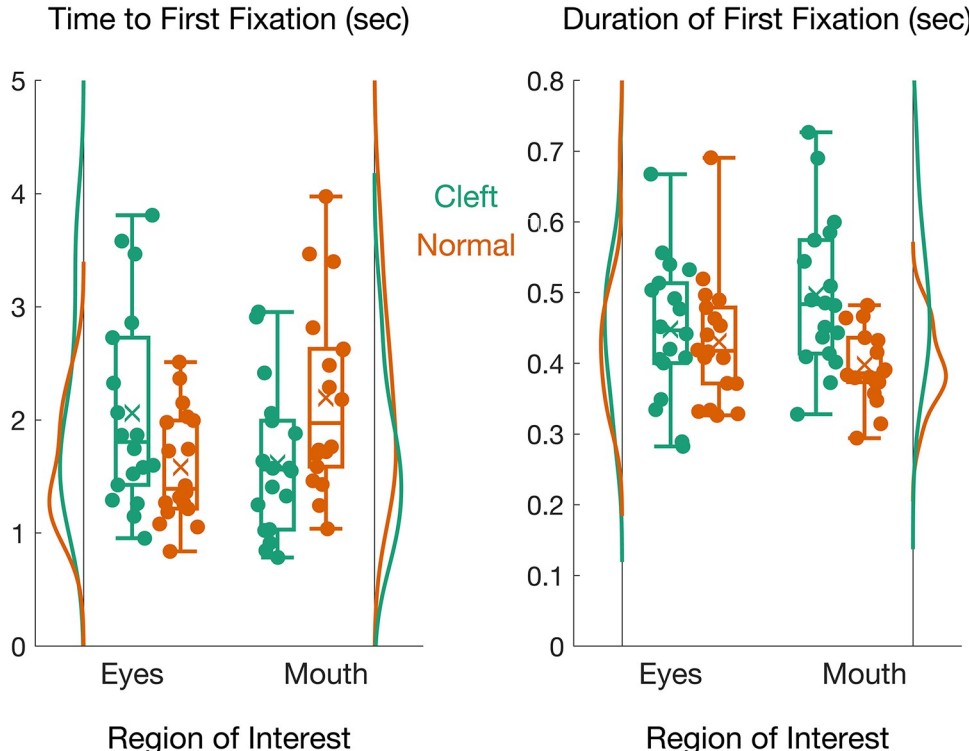

**Fig 4.** Vertical raincloud plots for the time to first fixation (left) and duration of first fixation (right).

**Time to the first fixation.** There was no main effect of *palate* ($F(1,17)$ = .24, $p$ = .634, $\eta^2$G = .001) or *ROI* ($F(1,17)$ = .10, $p$ = .757, $\eta^2$G = .004). However, there was a significant *palate* X *ROI* interaction ($F(1,17)$ = 35.51, p < .001, $\eta^2$G = .123; see Fig 4). Paired t-tests showed that the time to first fixation of the eyes was significantly slower ($t(17)$ = 3.72, p = .002, $M_{cleft}$ = 2.06 ± 0.88 sec, $M_{normal}$ = 1.58 ± 0.49 sec), while time to first fixation of the mouth was faster for the cleft palate than the normal palate faces ($t(17)$ = -4.02, p < 0.001, $M_{cleft}$ = 1.62 ± 0.65 sec, $M_{normal}$ = 2.20 ± 0.83 sec). This finding suggests that visual attention is more readily captured by the mouth region, at the expense of the eyes, for infant faces with cleft lip/palate.

**Duration of the first fixation.** There was a main effect of *palate* ($F(1,17)$ = 10.44, $p$ = .005, $\eta^2$G = .100) but no main effect of *ROI* ($F(1,17)$ = 0.38, $p$ = .545, $\eta^2$G = .002). Again there was a significant *palate* X *ROI* interaction ($F(1,17)$ = 8.69, $p$ = .009, $\eta^2$G = .052, see Fig 4). Paired t-tests showed that the interaction resulted because of the duration of the first fixation to the mouth ROI was significantly longer for cleft palates than normal palates ($t(17)$ = 4.59, p < 0.001, $M_{cleft}$ = 0.50 ± 0.11 sec, $M_{normal}$ = 0.40 ± 0.05 sec). There was no difference in the first fixation duration for the eye ROI between cleft and normal palate faces ($t(17)$ = 0.74, $p$ = .470, $M_{cleft}$ = 0.49 ± 0.10 sec, $M_{normal}$ = 0.43 ± 0.09 sec).

## EEG results

Repeated measures ANOVA was used to evaluate the effects of the within-subject factors of the palate (cleft/normal) and inversion (upright/inverted) on each ERP component. The hemisphere (L/R) acted as an additional within-subject factor for the N170. To account for the increase in family-wise error rate associated with running statistical tests on the four ERP measures, we set our pre-test alpha at .0125 (i.e., .05/4). All posthoc tests were Bonferroni corrected using the p.adjust feature in R.

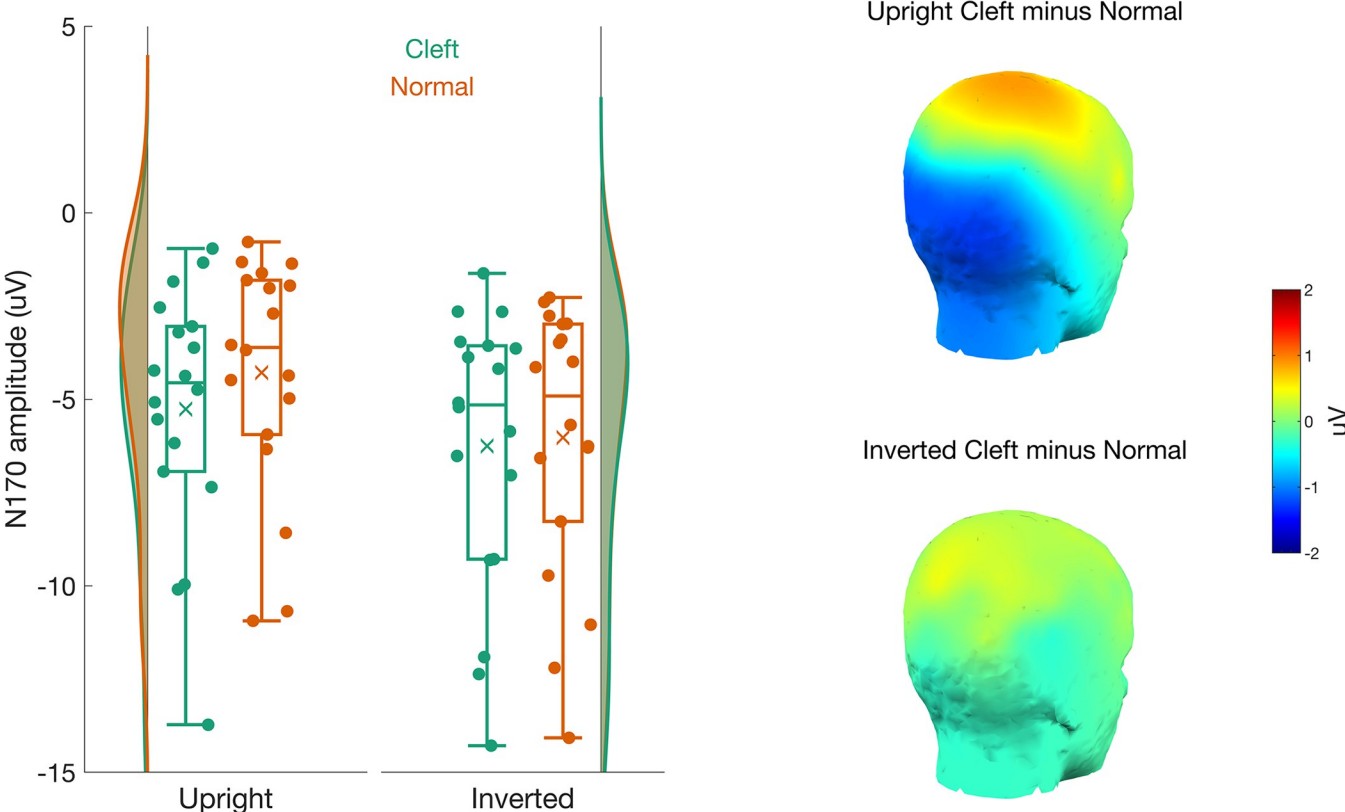

**Fig 5.** The left panel shows a vertical raincloud plot of the N170 amplitudes for the cleft (green) and normal (red) palate faces. The N170 amplitude for upright faces is on the left, and for inverted faces is on the right. *Note that the N170 gets larger as it gets more negative.* The 3D scalp topographies on the right highlight the source of the palate x inversion interaction. The N170 to cleft palate faces were much larger than to normal palate faces when displayed upright (top) but not when inverted (bottom).

**P100.** There was a main effect of *orientation* ($F(1,17) = 141.80$, $p < .001$, $\eta^2G = .104$), with the inverted faces (M = 4.63 ± 2.79 uV) generating a larger P100 than upright faces (M = 2.74 ± 2.87 uV). No other effects or interactions were significant at our adjusted alpha level (both $p > .01$, both $\eta^2G < .007$).

**N170.** The N170 was evaluated using a 3-way ANOVA with factors of *orientation*, *palate*, and *hemisphere*. There was a main effect of *orientation* ($F(1,17) = 53.38$, $p < .001$, $\eta^2G = .033$), a main effect of *palate* ($F(1,17) = 12.36$, $p = .003$, $\eta^2G = .006$), and a significant *orientation* x *palate* interaction ($F(1,17) = 8.40$, $p = .010$, $\eta^2G = .002$). We found no effects of *hemisphere* or interactions between *hemisphere* and the other factors (all $p > .17$, all $\eta^2G < .039$). Fig 5 shows the N170 amplitude collapsed across the left and right hemisphere. The *orientation* x *palate* interaction resulted because the cleft palate faces generated a larger N170 than the normal palate faces when upright ($t(17) = -3.69$, $p = .002$, $M_{cleft} = -5.07 ± 3.38$ uV, $M_{normal} = -4.00 ± 2.97$ uV) but not when inverted ($t(17) = -1.64$ $p = .119$, $M_{cleft} = -6.18 ± 3.76$ uV, $M_{normal} = -5.91 ± 3.55$ uV).

**P200.** There was a main effect of *orientation* ($F(1,17) = 28.09$, $p < .001$, $\eta^2G = .120$), a main effect of *palate* ($F(1,17) = 18.06$, $p < .001$, $\eta^2G = .017$), and a near-significant *orientation* x *palate* interaction ($F(1,17) = 8.06$, $p = .011$, $\eta^2G = .005$) on the amplitude of the P200. As shown in Fig 6, the interaction resulted because the P200 was larger for normal than cleft palates in the upright ($t(17) = -5.05$, $p < .001$, $M_{cleft} = 3.41 ± 2.61$ uV, $M_{normal} = 4.50 ± 2.46$ uV),

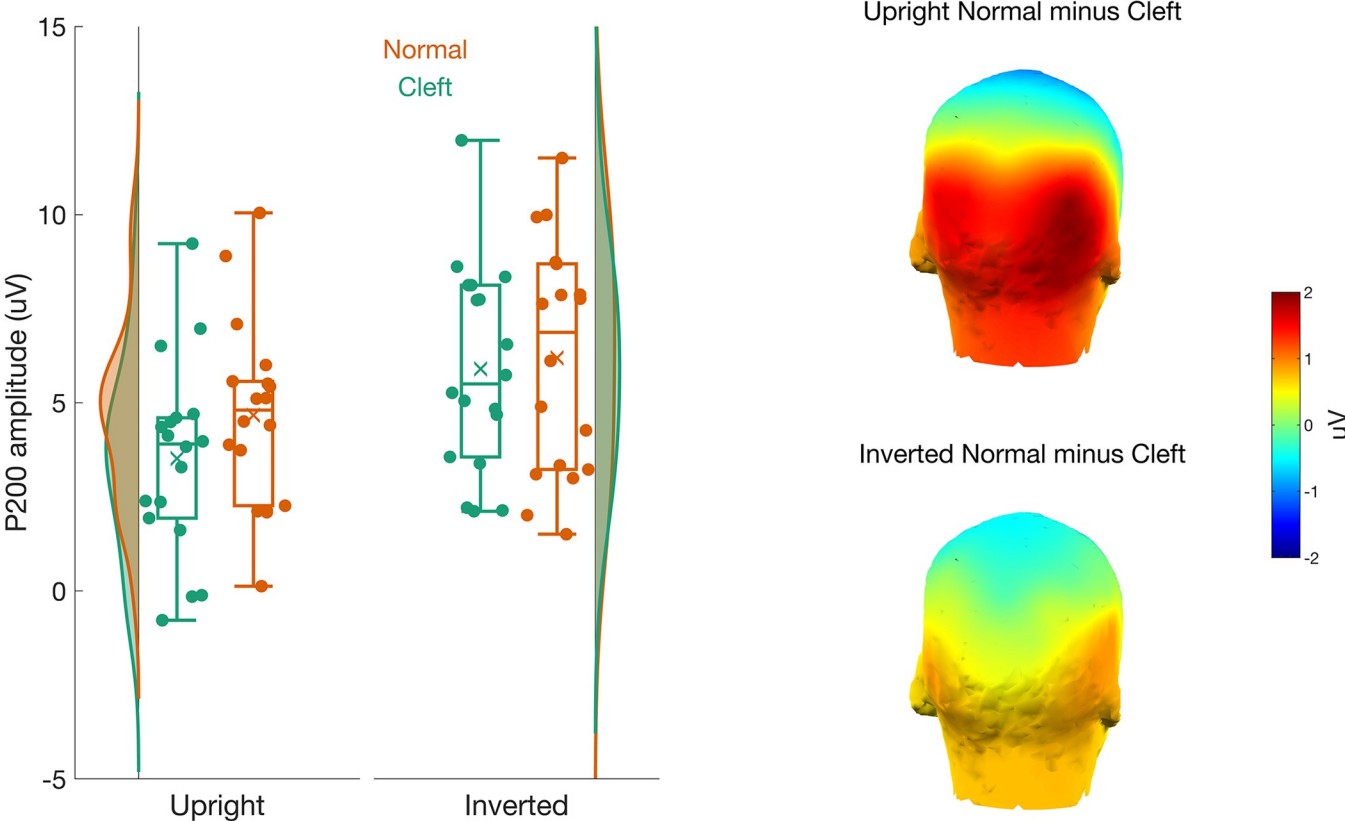

**Fig 6.** The left panel shows a vertical raincloud plot of the P200 amplitudes for the cleft (green) and normal (red) palate faces. The P200 amplitude for upright faces is on the left, and for inverted faces is on the right. The 3D scalp topographies on the right highlight the source of the palate x inversion interaction. The P200 to normal palate faces was much larger than to cleft palate faces when displayed upright (top) but not when inverted (bottom).

but not inverted condition (t(17) = -1.43, p = .171, $M_{cleft}$ = 5.78 ± 2.82 uV, $M_{normal}$ = 6.09 ± 3.07 uV).

**LPP.** There was a main effect of the *palate* ($F(1,17)$ = 11.66, $p$ = .003, $\eta^2 G$ = .034) such that the LPP was larger for cleft faces (M = 3.08 ± 1.63 uV) than normal faces (M = 2.51 ± 1.51 uV; see Fig 7). Although the main effect of *orientation* reached a nominal significance level ($F(1,17)$ = 6.12, $p$ = .024, $\eta^2 G$ = .038), it did not exceed our corrected error rate of .01. There was no significant interaction between *orientation* and *palate* ($F(1,17)$ = 0.147, $p$ = .706, $\eta^2 G$ < .001).

## Discussion

The current study utilized a perceptual measure (cuteness rating) along with both eye tracking and electroencephalography (ERPs) to investigate how cleft lip/palate impacts the early visual processing of infant faces. Our findings replicate previous work demonstrating that the disruption to normal face configuration posed by cleft lip/palate significantly affects gaze behavior [28] and partially replicates work showing that cleft lip/palate also disrupts early neural responses [24] linked to the perceptual encoding of faces. Importantly, our findings advance the current understanding of how cleft lip/palate may impact adults' responses to infants by demonstrating that the disruption to normal face configuration posed by cleft lip/palate also significantly affects later neural responses linked to affective processing.

In line with previous studies on perceptual responses to cleft lip/palate in humans [24, 26–28], we found that infants with a cleft lip/palate were rated as significantly less cute than those

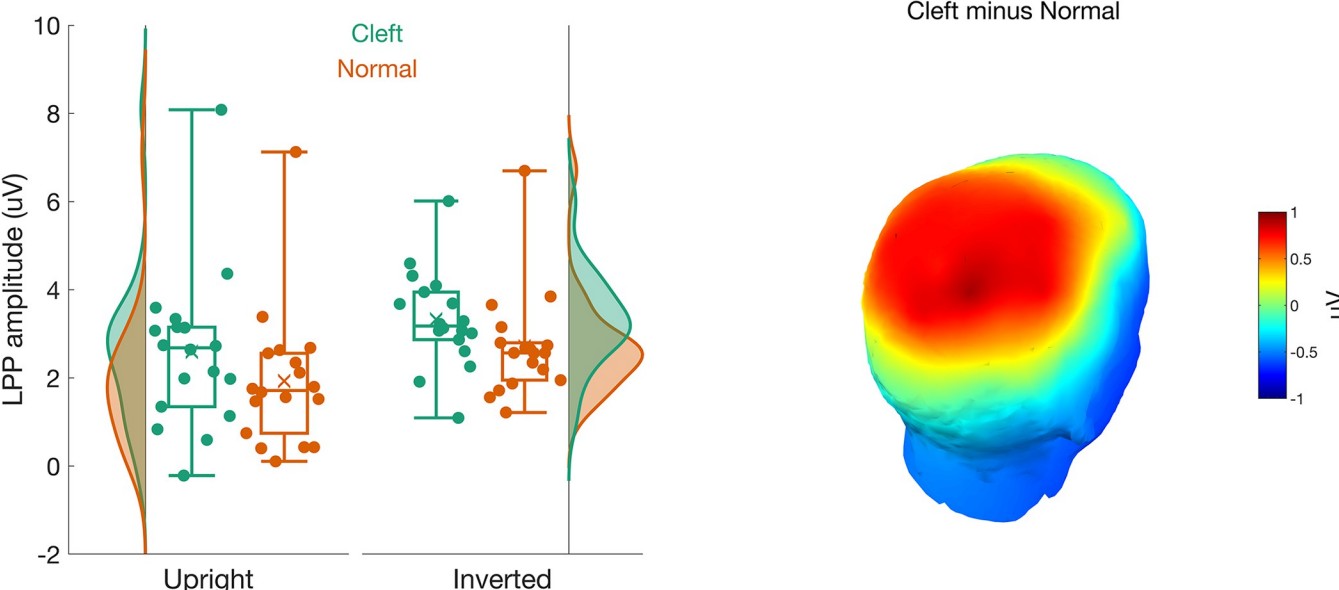

**Fig 7.** The left panel shows a vertical raincloud plot of the LPP amplitudes for the cleft (green) and normal (red) palate faces. The LPP amplitude for upright faces is on the left, and for inverted faces is on the right. The 3D scalp topographies on the right highlight the source of the effect of the palate. The LPP was larger for cleft than normal palate faces.

without. Although cuteness may seem trivial upon first reflection, it is actually one of the most power forces shaping our behavior [63]–it extends beyond culture [64, 65] and species [27, 65, 66]. Konrad Lorenz [29] proposed that the baby-typical features we perceive to be cute ('baby schema', e.g., large eyes, chubby cheeks, small chin, etc.; [67–72]) act as an innate releasing mechanism for parental behavior. Cuteness thus acts as a protective mechanism that ensures infant survival by triggering caretaking and defense for offspring that would otherwise be completely helpless. Indeed, adults routinely report being more willing to care for, protect, and form close bonds with infants displaying facial cues that are perceived to be cute [65, 67, 69, 73–75]. Cuteness also increases behavioral carefulness [76, 77]. The impact of perceived cuteness undoubtedly extends to long-term developmental outcomes; cleft lip/palate reducing the perceived cuteness of infants' faces is likely to have a range of negative impacts on their development. Particularly concerning is the link between cuteness and more negative responses. Low cuteness in infants/children has been linked to increased adult punitive responses [78] and more rejecting or non-supportive parenting behavior [79]. Cuteness even impacts actual medical outcomes–Badr-Zahr and Abdallah [6] found that premature infants in the NICU who were perceived to be less cute were less likely to thrive (measured by weight gain and length of hospital stay) than their cuter counterparts, which they argued could have been due to reduced nurturance from the healthcare providers.

Infant facial appearance plays a critical role in facilitating the caregiver-infant relationship. This relationship is dynamic in nature, relying on signals from both the infant and the caregiver. A particularly salient signal is that of eye gaze, which facilitates joint attention, social bonding, and empathy [5]. Our study demonstrated that cleft lip/palate structural abnormalities are associated with altered gaze patterns. Participants spent significantly longer viewing the mouth region for faces with cleft lip/palate than they did for those without. These findings align with previous work demonstrating that adults spent significantly more time viewing the mouth region of infant faces with cleft lip/palate compared to normal infant faces [28]. Our work extends these findings by demonstrating that not only do participants look longer at the

mouth region, but they are also quicker to direct their attention to the mouth for infants with cleft lip/palate at the cost of attentional prioritization of the eyes. Specifically, we found that the time to first fixation at each ROI varied as a function of cleft lip/palate; adults were quicker to fixate on the eye region for normal faces but quicker to fixate on the mouth for faces with cleft lip/palate. These findings suggest that the presence of cleft lip/palate may impact normative visual scanning patterns by drawing attention away from the eyes. This attentional shift may then negatively impact caretaking responses down the line as heightened visual attention to the disfigured area of faces has previously been linked to negative emotional experiences and stigmatizing affective responses [39].

In addition to altered gaze patterns, our study provides evidence for altered neural responses both early and late in face processing due to cleft lip/palate. Consistent with Huffmeijer et al. [24], we found reduced P200 responses as a function of cleft lip/palate. The P200 is generally thought to reflect the encoding of the second-order spatial relations of a face and is considered to be sensitive to the "typicality" of a face [43]. The finding that the P200 is smaller for faces with cleft lip/palate than normal faces may thus be a neural signature of the increased distance of faces with cleft lip/palate from typicality. This would be in keeping with studies showing a decrease in P200 with increasing distance of the morphed face from the typical or average face that the stark disruption of the facial configuration imposed by the cleft palate may represent [see 80]. That this pattern was apparent only in the upright orientation is expected, given that inversion disrupts the special relationship among facial features.

The current work provides support for the claim that the N170 component is affected by the configural disruption that occurs with cleft lip/palate [24]. Interestingly, Huffmeijer found attenuation of the N170 response to faces with cleft lip/palate as compared to normal faces. In contrast, the current study found enhanced N170 responses to faces with cleft lip/palate compared to normal faces. Although Huffmeijer suggest that the observed decreased N170 amplitude in response to cleft lip/palate indicates that the presence of a cleft lip/palate disrupts normative face processing, N170 amplitude increases are more commonly reported following configural disruptions such as facial inversion [e.g., 45, 81, 82], contrast reversal [e.g., 81], and Thatcherization [e.g., 83, 84]. Huffmeijer and colleagues suggest that N170 decreases in response to configural disruption may be related to the reduced processing requirements of passive viewing tasks. However, although ours was a passive viewing task, our participants still demonstrated enhanced N170 amplitudes. Differences in experimental factors may offer an alternative explanation. Stimuli presented by Huffmeijer were small (4 degrees of visual angle) relative to our images, which subtended a larger area of the retina (approximately 12 degrees). Although the impact of stimulus image size on the N170 has not been systematically investigated, a recent model for understanding holistic and featural processing of faces suggests that it is an important factor [85]. Substantive changes in the image size would change the retinal distance between critical facial features such as eyes and the cleft lip and, according to the lateral inhibition, face template and eye detector model [LIFTED, 85], could alter the impact of lateral inhibition between features represented at foveal and parafoveal regions.

Enhanced N170 responses to configural disruptions may reflect greater effort for so-called "face-specific" mechanisms due to difficulty extracting configural information [86, 87] and/or the recruitment of additional (not "face-specific") processing mechanisms early in visual processing [88]. Using a competition ERP paradigm, Sadeh and Yovel [49] provided evidence that rather than increase activity in already engaged face processing ensembles, the increase in N170 amplitude accompanying face inversion results in the recruitment of additional neural ensembles. This finding is consistent with fMRI research showing that face inversion results in decreased activity in face-specific regions and increased activity in more general object-processing regions [see 89 for a review]. These findings are consistent with the explanation that

face manipulations such as inversion disrupt configural processing in face-specific areas and promote the recruitment of generic object processing mechanisms that aid in identifying and understanding the altered facial features. In the current case, the cleft palate/lip may distort the configuration of the face sufficiently to engage other non-face processing mechanisms. Interestingly, recent work has suggested that inversion impacts cuteness perception in a task-dependent fashion [90]; it remains unknown whether infant face processing relies on configural processing to the same degree as adult face processing. Although disruptions of the N170 represent early perceptual stages of processing, evidence suggests it can potentially lead to dehumanizing responses. Hugenberg et al. [91] demonstrated that disruptions of face configuration by inversion had several social impacts, including a reduction in the ability of a human face to activate concepts related to "humanness," delay the categorization of a face as human, and reduce the levels of humanlike traits ascribed to faces. Taken together, this literature suggests that differences in early perceptual mechanisms can contribute to the negative impact of a cleft lip/palate on infant welfare.

Importantly, the increase in LPP for both upright and inverted cleft faces provides evidence that the impact of cleft lip/palate on neural processing extends to late-stage affective face processing. The LPP is thought to reflect automatic orienting (Hajcak et al., 2009) and sustained allocation of attention [50, 92, 93] to emotionally or motivationally relevant stimuli. Some researchers have argued that enhanced LPP responses may be indicative of more elaborate or amplified stimulus processing [94, 95] or the inhibition of possibly competing visual representations [96]. Enhanced LPP responses to infant faces with cleft lip/palate relative to normal infant faces may indicate the potential emotional salience of the face and heighten visual processing. Although both positive and negative affective stimuli can induce an LPP, the decreased cuteness ratings observed during the eye tracking portion of this study suggest that the increased LPP reflects a negative affective state in participants. Trial-by-trial analysis simultaneous EEG and fMRI show a correlation between the LPP to negative stimuli and activation in the anterior insula [51], a brain region associated with integrating cognitive, emotional, and sensory processes to generate subjective feelings and awareness and help guide future actions and decisions. Within this framework, the increased LPP to infants' faces may signal a behavior change away from the typical caregiving response typical when viewing infants' faces.

While the current study provides novel insight into the impact cleft lip/palate may have on adults' visual perception of infants, there are several limitations of the current work worth mentioning. Firstly, this study relies on a relatively small sample. Although smaller samples are not at all uncommon in EEG studies, with the average sample size being N = 21 [97, 98], it is nonetheless a limitation in the current work. Data collection for this study occurred during the COVID-19 pandemic, which created difficulty recruiting participants for an in-lab study. Additionally, this study compares responses to infants with cleft lip/palate and healthy infant faces using images of different infant identities. Ideally, we would be able to compare perceptual and neural responses to the same faces with and without cleft lip/palate, however this is not possible without the use of image manipulation (e.g., photoshop), which may alter the realism of the cleft lip/palate on the face. One such study [99] did provide some evidence that scarring following craniofacial repair may impact visual scanning as compared to images that were photoshopped to remove scarring. Future work should explore this issue in more depth.

Additionally, it is worth noting that this study was conducted on a non-parent sample. The use of nulliparous individuals is common in studies exploring adults' neural and behavioral responses to infants, however [e.g., 8, 27, 30, 65, 75, 100–109]. Infant faces are highly biologically-relevant stimuli that have been found to capture adult attention [110] and elicit positive affective responses [111] in both parents and non-parents. For example, Thompson Booth and colleagues [110] found that infant faces preferentially engage visual attention relative to adult

faces in both mothers and non-mothers (although this effect was stronger in mothers). Neuroimaging work has similarly demonstrated that infant cues (both facial and vocal) capture attention and elicit reward-related neural activity (relative to adult cues) even among nonparents [reviewed in 112]. Results from ERP studies, however, are more mixed–some studies have found differences in the N170 [113] and LPP [114] responses to infant faces among parents compared to non-parents, while others have not [115, 116]. A recent meta-analysis indicates that these parental status effects may hold true for the N170 (with slightly larger N170 responses among parents compared to non-parents) but not the LPP [117]. While it is possible that parental status may impact the responses observed here, Schaller [31] has recently argued that humans have evolved a *parental care motivation system* reflecting psychological mechanisms that have evolved to promote caretaking behavior and parental "instincts" in parents and non-parents alike. Future studies may explore the impact of parental status on adults' responses to cleft lip/palate (or other facial malformations) to more directly address the impact these have on parent-child interactions.

## Conclusions

The current findings provide insight into the multifaceted impacts of cleft lip/palate on adults' responses to infant faces that may impact social interactions and bonding, revealing that cleft lip/palate influences visual attention, disrupts the configural processing of faces, and heightens emotional processing. These results highlight the potential challenges infants with cleft lip/palate face in establishing and maintaining social bonds due to alterations in fundamental aspects of face perception. Firstly, cleft lip/palate reduces eye contact which is crucial for social communication and bonding. The prioritization of the affected mouth region over the eyes for infants with cleft lip/palate may contribute to a myriad of negative outcomes regarding maternal closeness, bonding, and social engagement. Secondly, cleft lip/palate disrupts the holistic processing (as evidenced by the N170) and emotional processing (as evidenced by the LPP) of infant faces, which could reduce the release of innate caretaking behaviors associated with typical infant features [29, 63]. These findings contribute to a better understanding of the complex interplay between facial atypicality, neural processing, and social functioning in individuals with cleft lip/palate. Moreover, our results emphasize the need for increased awareness, support, and targeted interventions to counteract potential negative consequences and ensure that all children, regardless of their facial appearance, receive the nurturing care they need to thrive. For example, many infants with cleft lip/palate undergo craniofacial surgery to correct the facial malformation, the timing of which has been shown to impact mother interactions [19]. It remains unclear, however, if/how surgical repair of the cleft lip/palate may impact or restore visual and neural processing of infants' faces. Future work should seek to expand these findings to explore this critical issue.

## Acknowledgments

The authors would like to thank the dedicated team of student research assistants in the Behavioral Endocrinology Research Lab who assisted with data collection.

## Author Contributions

**Conceptualization:** Amanda C. Hahn, Kelly J. Jantzen.

**Formal analysis:** Amanda C. Hahn, Kelly J. Jantzen.

**Investigation:** Amanda C. Hahn, Zoë Royer, Jeffrey Frederick, Rachael Kee, Rhiannon Crimmins, David H. Harris, Kelly J. Jantzen.

**Methodology:** Amanda C. Hahn, Kelly J. Jantzen.

**Resources:** Amanda C. Hahn, Kelly J. Jantzen.

**Software:** Kelly J. Jantzen.

**Supervision:** Amanda C. Hahn, Kelly J. Jantzen.

**Visualization:** Amanda C. Hahn, Kelly J. Jantzen.

**Writing – original draft:** Amanda C. Hahn, Kelly J. Jantzen.

**Writing – review & editing:** Amanda C. Hahn, Juergen A. Riedelsheimer, Bernd Huber, Kelly J. Jantzen.

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
