## [Decision Letter · Decision Letter 0]

27 Dec 2023

PONE-D-23-31387Effects of Cleft Lip on Visual Scanning and Neural Processing of Infant FacesPLOS ONE

Dear Dr. Hahn,

Thank you for submitting your manuscript to PLOS ONE. After careful consideration, we feel that it has merit but does not fully meet PLOS ONE’s publication criteria as it currently stands. Therefore, we invite you to submit a revised version of the manuscript that addresses the points raised during the review process.

It is recommended that you address each of the reviewers comments in your revision paying particular attention to the concerns about lack of explanation of methodology that was implemented. Several variables and approaches require further exploration and explanation.

We look forward to receiving your revised manuscript.

Kind regards,

JJ Cray Jr., Ph.D.

Academic Editor

PLOS ONE

3. We note that Figures 1, 5, 6, and 7 in your submission contain copyrighted images. All PLOS content is published under the Creative Commons Attribution License (CC BY 4.0), which means that the manuscript, images, and Supporting Information files will be freely available online, and any third party is permitted to access, download, copy, distribute, and use these materials in any way, even commercially, with proper attribution. For more information, see our copyright guidelines: http://journals.plos.org/plosone/s/licenses-and-copyright.

1. You may seek permission from the original copyright holder of Figures 1, 5, 6, and 7 to publish the content specifically under the CC BY 4.0 license.

Reviewers' comments:

Reviewer's Responses to Questions

**Comments to the Author**

1. Is the manuscript technically sound, and do the data support the conclusions?

Reviewer #1: Yes

Reviewer #2: Yes

2. Has the statistical analysis been performed appropriately and rigorously? 

Reviewer #1: Yes

Reviewer #2: Yes

3. Have the authors made all data underlying the findings in their manuscript fully available?

Reviewer #1: Yes

Reviewer #2: Yes

4. Is the manuscript presented in an intelligible fashion and written in standard English?

Reviewer #1: Yes

Reviewer #2: Yes

5. Review Comments to the Author

Reviewer #1: This paper reports on a single study (N = 18) in which subjective ratings of cuteness, gaze behavior, and event-related potentials (ERPs) were measured in response to infant faces with and without cleft lip/palate. The results showed that these measures were affected by the presence of cleft lip/palate. Most of the findings, with the exception of N170 amplitude, were consistent with those of previous studies. Specifically, infant faces with cleft lip/palate were rated as cuter, their eye regions were viewed less intensely, and they produced enhanced ERP responses compared to infant faces without cleft lip/palate.

This paper contributes some findings to the literature. The small sample size is a weakness of this study, yet the findings seem reasonable and robust. Below I will mention some points that should be elaborated before its publication.

Major concerns.

1. Stimulus characteristics

As the authors noted in the Discussion, the lack of stimulus control is a weakness of this study. Although the use of realistic stimuli has some merit, both gaze and ERP measures are sensitive to the physical properties of stimuli. For example, Figure 1 shows that the face of an infant with cleft lip/palate may have different properties in the eye region than the face of a normal infant. This possibility can be clearly stated as a limitation of the study. In this context, please describe the details of the stimuli more concretely, such as size and brightness. Were they presented at the same size for the eye tracking and ERP experiments? Was the chin rest also used for the ERP experiment to keep the viewing distance constant?

2. The definition of cuteness

Please define the cuteness of infant faces more precisely in the introduction. There was virtually no explanation in the current form. Part of the discussion of the meaning of cuteness can be moved to the Introduction. In this context, some of the previous studies cited in the Discussion dealt with "attractiveness" rather than "cuteness" (e.g., lines 347–350). Although I understand that cuteness is almost synonymous with infant attractiveness in the current context, a careful description would be preferable.

3. Inverted face condition

The inverted face condition was only included in the ERP experiment. I think that including it in the eye tracking and subjective rating experiments would have broadened the discussion. For subjective ratings, a recent study showed that the cuteness of infants' faces is not affected by face inversion, possibly because the perception of cuteness is mainly based on elemental features such as roundness (Kuraguchi & Nittono, 2023, Perception, https://doi.org/10.1177/03010066231198417). What would be expected in the visual scanning experiment when infant faces with cleft lip/palate are presented upside down?

4. Eye tracking results

First, please write the unit of measurement for eye tracking and ERP measures in the main text. When the unit of measurement for eye tracking is seconds, the mean values seem strange. It is unlikely that it took more than 1.5 seconds to fixate on the eye position of the faces of normal infants (lines 254-256). Furthermore, the duration of the first fixation is said to be less than 1 second (lines 264-266). Does the duration mean literally the first fixation or the dwell time on the ROI? Please specify.

Minor comments.

1. Sample Size

Please provide a rationale for the current sample size in the Methods section.

2. Sample characteristics

In the present study, young non-parents served as participants. I don't think this sample selection is optimal to discuss the mother (or father)–infant relationship. At least a comment on its effect on the generalizability of the results would be worth adding to the Discussion.

3. Eye tracking experiment

The authors wrote that the stimulus duration was 10 s. What were the interstimulus intervals? Please provide more explanation, possibly in the Methods section, as to why the authors focused on the two measures, time to first fixation and duration of first fixation, and the more concrete operational definitions of them. It seems unlikely that it took more than 1.5 seconds to fixate on the eye region of the faces of normal infants (lines 254–256). Does the duration of the first fixation mean literally the first single fixation or the dwell time on the ROI? Please specify.

4. ERP experiment

Please describe the instructions to the participant more concretely in the Method section.

Typos.

line 213. Pion et al. should read Pion-Tonachini et al.

line 228. P03, P0z, P04 should be PO3, POz, PO4 (not zero but O).

line 312. error rate of .001 should be error rate of .01.

Reviewer #2: This is an interesting and well-designed study. I have attached my comments as a word document.

While very clear for the most part, there are bits that require editing to improve clarity. One question I have is on the images, e.g. the face used as an example in the paper - are there any ethical issues around publishing this image?

6. PLOS authors have the option to publish the peer review history of their article (what does this mean?). If published, this will include your full peer review and any attached files.

Reviewer #1: No

Reviewer #2: No

---

## [Author Response · Author response to Decision Letter 0]

31 Jan 2024

Please see attached response to reviewers document

---

## [Decision Letter · Decision Letter 1]

28 Feb 2024

PONE-D-23-31387R1Effects of Cleft Lip on Visual Scanning and Neural Processing of Infant FacesPLOS ONE

Dear Dr. Hahn,

Thank you for submitting your manuscript to PLOS ONE. After careful consideration, we feel that it has merit but does not fully meet PLOS ONE’s publication criteria as it currently stands. Therefore, we invite you to submit a revised version of the manuscript that addresses the points raised during the review process.

There are a few minor editorial issues to be addressed before acceptance.

We look forward to receiving your revised manuscript.

Kind regards,

JJ Cray Jr., Ph.D.

Academic Editor

PLOS ONE

Journal Requirements:

Reviewers' comments:

Reviewer's Responses to Questions

**Comments to the Author**

1. If the authors have adequately addressed your comments raised in a previous round of review and you feel that this manuscript is now acceptable for publication, you may indicate that here to bypass the “Comments to the Author” section, enter your conflict of interest statement in the “Confidential to Editor” section, and submit your "Accept" recommendation.

Reviewer #1: All comments have been addressed

Reviewer #2: All comments have been addressed

2. Is the manuscript technically sound, and do the data support the conclusions?

Reviewer #1: Yes

Reviewer #2: Yes

3. Has the statistical analysis been performed appropriately and rigorously? 

Reviewer #1: Yes

Reviewer #2: Yes

4. Have the authors made all data underlying the findings in their manuscript fully available?

Reviewer #1: Yes

Reviewer #2: Yes

5. Is the manuscript presented in an intelligible fashion and written in standard English?

Reviewer #1: Yes

Reviewer #2: Yes

6. Review Comments to the Author

Reviewer #1: I think the authors have answered all the points I raised. I would be happy to support its publication after the authors address the following minor comments.

Lines 290-291. "Time to first fixation was defined as the onset of the first gaze fixation in each ROI."

Please specify that it was defined as the interval between the stimulus onset to the first gaze onset.

Lines 296-297. "To account for the increase in family-wise error rate associated with running multiple statistical tests, we set our pre-test alpha at .025 (i.e., 297 .05/2)."

Lines 329-330. "To account for the increase in family-wise error rate associated with running multiple statistical tests, we set our pre-test alpha at .0125 (i.e., .05/4)."

Please consider add "on the two measures" and "on the four ERP components," respectively, after "multiple statistical tests" for better understanding. In fact, multiple statistical tests can mean multiple tests for main and interaction effects in a single multifactorial ANOVA.

Reviewer #2: (No Response)

7. PLOS authors have the option to publish the peer review history of their article (what does this mean?). If published, this will include your full peer review and any attached files.

Reviewer #1: No

Reviewer #2: No

---

## [Author Response · Author response to Decision Letter 1]

29 Feb 2024

Please see attached response document

---

## [Editor Report · Decision Letter 2]

4 Mar 2024

Effects of Cleft Lip on Visual Scanning and Neural Processing of Infant Faces

PONE-D-23-31387R2

Dear Dr. Hahn,

We’re pleased to inform you that your manuscript has been judged scientifically suitable for publication and will be formally accepted for publication once it meets all outstanding technical requirements.

Kind regards,

JJ Cray Jr., Ph.D.

Academic Editor

PLOS ONE
---

## [Editor Report · Acceptance letter]

18 Mar 2024

PONE-D-23-31387R2 

PLOS ONE

Dear Dr. Hahn, 

I'm pleased to inform you that your manuscript has been deemed suitable for publication in PLOS ONE. Congratulations! Your manuscript is now being handed over to our production team.

Kind regards, 

on behalf of

Dr. JJ Cray Jr. 

Academic Editor

PLOS ONE